# Assessment of knowledge, practice, and status of food handlers toward Salmonella, Shigella, and intestinal parasites: A cross-sectional study in Tigrai prison centers, Ethiopia

**Fitsum Mardu**[1]*, **Hadush Negash**[2], **Haftom Legese**[2], **Brhane Berhe**[1], **Kebede Tesfay**[1], **Hagos Haileslasie**[3], **Brhane Tesfanchal**[4], **Gebremedhin Gebremichail**[3], **Getachew Belay**[4], **Haftay Gebremedhin**[5]

1 Unit of Medical Parasitology, Department of Medical Laboratory Sciences, College of Medicine and Health Sciences, Adigrat University, Adigrat, Ethiopia, 2 Unit of Medical Microbiology, Department of Medical Laboratory Sciences, College of Medicine and Health Sciences, Adigrat University, Adigrat, Ethiopia, 3 Unit of Hematology and Immunohematology, Department of Medical Laboratory Sciences, College of Medicine and Health Sciences, Adigrat University, Adigrat, Ethiopia, 4 Unit of Clinical Chemistry, Department of Medical Laboratory Sciences, College of Medicine and Health Sciences, Adigrat University, Adigrat, Ethiopia, 5 Department of Public Health, College of Medicine and Health Sciences, Adigrat University, Adigrat, Ethiopia

* fmardu25@gmail.com

## Abstract

### Background

Unsafe food becomes a global public health and economic threat to humans. The health status, personal hygiene, knowledge, and practice of food handlers have crucial impact on food contamination. Hence, this study is aimed at assessing the knowledge, practice, and prevalence of Salmonella, Shigella, and intestinal parasites among food handlers in Eastern Tigrai prison centers, Northern Ethiopia.

### Methods

An institutional-based cross-sectional study was carried out from April to September 2019 among food handlers in Eastern Tigrai prison centers, Northern Ethiopia. A structured questionnaire was used to collect the demographic characteristics, the knowledge, and the practice of the study participants. Direct wet mount and formol-ether concentration techniques were applied to identify intestinal parasites. Culture and biochemical tests were used to isolate the Salmonella and the Shigella species. Additionally, antimicrobial susceptibility tests to selected antibiotics were performed using Kirby-Baur disk diffusion method. We used SPSS version 23 software for statistical analysis.

### Results

Thirty-seven (62.7%, 37/59) of the participants had harbored one or more intestinal parasites. The protozoan *Entamoeba histolytica/dispar* was detected among 23.7% (14/59) of

---

**Funding:** FM received fund from Adigrat University College of Medicine and Health Sciences, Adigrat, Ethiopia (Grant number AGU/CMHS/035/10) The funders had no role in study design, data collection and analysis, decision to publish, or preparation of the manuscript.

**Competing interests:** The authors have declared that no competing interests exist.

the study participants who provided stool specimen. Besides, 6.8% (4/59) of the samples were positive for either Salmonella or Shigella species. The Salmonella isolates (n = 2) were sensitive to Gentamicin, Ciprofloxacin, Ceftriaxone, and Clarithromycin but resistant to Amoxicillin, Ampicillin, and Amoxicillin/clavulanic acid. Similarly, the two Shigella isolates were susceptible to Gentamicin, Ciprofloxacin, and Ceftriaxone but showed resistance to Amoxicillin, Tetracycline, and Chloramphenicol. Further, 60.6% (40/66) of the participants had good level of knowledge, and 51.5% (34/66) had good level of practice on foodborne diseases and on food safety.

## Conclusions

We conclude that foodborne pathogens are significant health problems in the study areas. Regular health education and training programs among the food handlers are demanded to tackle foodborne diseases at the prison centers.

## Background

Unsafe food becomes a global public health and economic threat to humans. According to WHO, more than 600 million illnesses and 420,000 annual deaths worldwide are due to contaminated food. In addition, about 33 million Disability Adjusted Life Years (DALYs) are attributable to foodborne infections globally [1]. People in low- and middle-income countries are at high risk of foodborne diseases related to poor sanitation and lack of food safety practices [2]. In these countries, foodborne diseases cost more than US$ 110 billion losses in human productivity each year [3].

The highest burden of foodborne disease mortality has been reported in Africa, where diarrheal disease agents are the leading causes [1, 4]. In Ethiopia, deaths due to diarrheal diseases have reached up to 54,357 (8.55% of the total deaths), which has ranked the country 20th in the world [5]. Moreover, the annual incidence of foodborne diseases ranges from 3.4% to 9.3%, with associated deaths ranging from 22.6% to 62% [6]. Despite the evident importance of foodborne diseases, Ethiopia is among the countries that does not have a prioritized food safety policy [7].

Salmonella and Shigella are among the common causes of foodborne diseases throughout the world [8]. Salmonella species cause over 25 million annual incidences, with more than 200,000 associated deaths. The estimated global number of incidences of Shigella species is 165 million [9]. Numerous helminths and protozoa are also transmitted by contaminated food especially when food service workers have poor personal hygiene or work in unsanitary situations [10].

Food handlers have an important role in the spread of foodborne diseases in a community. They asymptomatically harbor foodborne pathogens leading to difficulties in the prevention and control of foodborne infections [11, 12]. Not only the health status and the personal hygiene of food handlers but also their knowledge and practice on food safety has a crucial impact on food safety [13]. Nevertheless, not all food handlers realize their roles in protecting their health and the health of the community from foodborne diseases [14, 15].

Assessing the knowledge, practice, and occurrence of foodborne pathogens among food handlers is important in implementing the prevention and control strategies of foodborne diseases. Thus, this study is aimed at assessing the knowledge and practice on food safety as well

as the prevalence of Salmonella, Shigella, and intestinal parasites among food handlers in Eastern Tigrai prison centers, Northern Ethiopia.

## Methods

### Study design, area, and period

We conducted an institutional-based cross-sectional study among food handlers in Eastern Tigrai prison centers, Northern Ethiopia from April to September 2019. According to the Central Statistics Agency of Ethiopia (2012), the Eastern zone of Tigrai has a total population of 862,348. The zone is located 900 kms north of Addis Ababa (the capital of Ethiopia) at a longitude and latitude of 14˚ 16′ N 39˚ 27′ E, with an elevation of 2457 m above sea level [16]. There are two zonal prison centers in Eastern Tigrai: Adigrat and Wukro prison centers. During the study period, the two prison centers had accommodated 2080 prisoners.

### Study participants

Our study population was all individuals engaged in food handling in Adigrat and Wukro prison centers during the study period. In this study, we enrolled two categories of food handlers: i) individuals from outside of the prison population who were involved in food preparation and ii) prisoners who were distributing prepared food to the whole prison population. As a result, sixty-six individuals (comprising 40 males) were working in food handling (preparation or distribution) in the study areas.

### Inclusion criteria

All food handlers (66) working at Adigrat and Wukro prison centers were eligible for participation.

### Sample size and sampling technique

Since the number of food handlers in the study areas was very small, we enrolled all of them in the study. Therefore, the total sample size for this study was 66. We applied total population sampling technique to enroll the study participants.

### Data collection methods and tools

Data regarding the socio-demographics, the knowledge, and the practice of the participants were collected by face-to-face interviews using a structured questionnaire (S1 File). The questionnaire was adapted from previous literature [12, 17]. The questions need 'Yes', 'No', or 'do not know' responses. Each correct response scores a value of '1' while an incorrect response has a 'zero' value. The cut-off points for a 'good' level of knowledge or a 'good' level of practice was the correct responses to >50% of the questions. We also described and discussed the responses of the participants to the individual questions.

### Specimen collection, processing, and examination

**Parasite identification.** The study participants were instructed to bring about 3 grams of stool specimen with clean dry container. Each sample was immediately examined using physiological (0.85%) saline and Lugol's iodine to detect intestinal protozoa. We then preserved the remained samples in a 10% (V/V) formalin solution (for concentration of parasites) and in Amies transport medium (for culture and biochemical testing). A Formol-ether concentration technique was performed to enhance the identification of intestinal parasites [18]. All the

laboratory procedures have been performed at Adigrat University, Ethiopia. (**See the protocol in** S1 File **for more details about the laboratory procedures**).

**Isolation of Shigella and Salmonella species.**    We cultured the stool samples in Salmonella-Shigella Agar (SS Agar) to identify the presumptive Salmonella and Shigella species. After inoculation and overnight incubation at 35˚c, the media were inspected for the growth of the suspected bacteria. Pure colonies with a characteristic of Salmonella- and/or Shigella-like species were further inoculated into selected biochemical tests namely Triple Sugar Iron Agar (TSI), Motility-Indole-Ornithine Agar (MIO), Urea test, and Simmons Citrate Agar for confirmation. Finally, the results of the biochemical profiles consistent with Salmonella and/or Shigella species were reported according to the standard protocol. (**See the protocol in** S1 File **for more details about the laboratory procedures**).

## Antimicrobial susceptibility testing

Antimicrobial susceptibility testing was performed using Kirby-Baur disk diffusion method [19] to identify which antimicrobial regimen was effective for each infected participant. After choosing well-isolated colonies from the positive culture, McFarland standard bacterial suspensions (inoculums) were prepared and inoculated on Mueller Hinton Agar. Then following the placement of antimicrobial disks using sterile forceps, the plates were incubated at 35˚c for 18 hours. Finally, applying the guideline of the Clinical Laboratory Standards Institute (CLSI) 2016 [20], we interpreted the results of the disks as 'susceptible' or 'resistant' by measuring the inhibition zones with a ruler. The following antibacterial drugs were tested for the Salmonella and Shigella isolates- Ampicillin (30 μg), Gentamicin (10 μg), Chloramphenicol (30 μg), Ciprofloxacin (5 μg), Tetracycline (10 μg), Ceftriaxone (30 μg), Amoxicillin-clavulanic acid (30 μg), Clarithromycin (30 μg), and Amoxicillin (30 μg).

## Quality control

To ensure the quality of data, the data collectors were trained and the questionnaire was pre-tested before the data collection. The collected data have daily been assessed for consistency and accuracy. The participants were oriented on proper sample collection. Besides, we checked the expiry date of the reagents before use. The quality of the culture media was checked by inoculating known strains of Salmonella (ATCC14028) and Shigella (ATCC23354) species. The temperatures of the incubator and the refrigerator were regularly being monitored. All the laboratory procedures were conducted as per the standard operating procedures.

## Statistical analysis

SPSS version 23 software was used for data analysis. Frequency distributions and percentages were computed for categorical variables. Bivariate logistic regression was applied to determine the crude association (using crude odds ratio) between the socio-demographic variables and the occurrence of Salmonella, Shigella, or intestinal parasites among the study participants. Variables with $p < = 0.2$ in the bivariate logistic regression were transferred to multivariate regression analysis to identify factors that have statistical significance with the presence of Salmonella, Shigella, or intestinal parasitic infections. A p-value less than 0.05 at 95% confidence level was considered as statistically significant association.

## Operational definition

**Multidrug-resistant isolates.**    Isolates which are resistant to antibiotic agents in at least 3 antimicrobial classes [21].

### Ethics approval and consent to participate

The study was approved by the Institutional Review Board of Tigrai Health Research Institute (**THRI/4031/0393/2018**). We then granted support letter from the Research and Community Service Directorate of Adigrat University, Ethiopia (**R-CS-D/10/03/2018**). In addition, we obtained official permission from the respective prison center administrations to conduct the study. More importantly, each study participant gave informed written consent to participate (consent form is provided in S1 File). The study participants had the opportunity to withdrew from the study at any time. Any information pertaining to participants has been kept confidential. Moreover, physicians at the prison centers have treated the infected study participants.

## Results

### Socio-demographics of the study participants

In this study, 66 food handlers were interviewed, of which 59 gave stool specimens. The age of the total participants ranged from 17 to 56 years; the mean was 25.42 years (SD 9.5). The majority (60.6%) of these were males. Also, forty-nine (74.2%) of the total participants were aged between 17 to 26 years. None of the socio-demographic variables were shown to be statistically associated with the occurrence of the target microorganisms among the participants (p >0.05) (Table 1).

### Prevalence of intestinal parasites among the participants

Of the total 66 volunteered participants, 59 (89.4%) gave stool specimens. Thirty-seven (62.7%, 37/59) of the total samples examined were positive for at least one intestinal parasite. The species *E. histolytica/dispar* was detected among 23.7% (14/59) of the samples examined. The double infections of *E. histolytica/dispar* and *G. lamblia* were identified from 17% of the samples. Conversely, no intestinal helminth parasite was detected among the participants of this study (Fig 1).

### Salmonella and Shigella isolates among the participants

From the 59 samples cultured on SSA, two Salmonella-like and two Shigella-like species were presumably identified. These were further confirmed to be Salmonella or Shigella species by inoculating in to selected biochemical tests. The overall combined prevalence of Salmonella and Shigella isolates was 6.8% (4/59). None of the samples harbored both Salmonella and Shigella isolates.

### Antimicrobial susceptibility of the Salmonella and the Shigella isolates

Both the Salmonella isolates were sensitive to Gentamicin, Ciprofloxacin, Ceftriaxone, and Clarithromycin. To the contrary, both the Salmonella isolates were resistant to Amoxicillin, Ampicillin, and Amoxicillin/clavulanic acid. Similarly, both the Shigella isolates showed susceptibility to Gentamicin, Ciprofloxacin, and Ceftriaxone; but were resistant to Amoxicillin, Tetracycline, and Chloramphenicol. Disturbingly, both of the Shigella isolates were multidrug-resistant (defined as resistance to at least three classes of antibiotics) (Table 2).

### Knowledge and practice of the study participants

Table 3 summarizes the knowledge and the practice of the participants on foodborne diseases and on food safety. In this study, 51.5% (34/66) of the participants interviewed had good food safety practices. Only 31 (47%) of the participants responded that they always wear gown while

**Table 1. Socio-demographic characteristics and the distribution of Salmonella, Shigella, and intestinal parasites among the participants at Adigrat and Wukro prison centers, Tigrai, Northern Ethiopia, 2019.**

| Variables | Frequency (n, %), $N_1$ = 66 | Presence of Salmonella, Shigella, or intestinal parasites ($N_2$ = 59) | | COR (95% CI) | p-value | AOR, (95% CI) | p-value |
|---|---|---|---|---|---|---|---|
| | | Yes (n, %) | No (n, %) | | | | |
| **Gender** | | | | | | | |
| Male | 40 (60.6) | 20 (60.6) | 13 (39.4) | 1.7 (0.6–5.3) | 0.3 | 2.2(0.7–7.3) | 0.17 |
| Female | 26 (39.4) | 19 (73.1) | 7 (26.9) | 1 | | | |
| **Age in years** | | | | | | | |
| 17–26 | 49 (74.2) | 31 (72.1) | 12 (27.9) | 0.7 (0.1–4.7) | 0.7 | - | - |
| 27–36 | 11 (16.7) | 4 (40) | 6 (60) | 3 (0.4–24) | 0.3 | | |
| > = 37 | 6 (9.1) | 4 (66.7) | 2 (33.3) | 1 | | | |
| **Education level** | | | | | | | |
| Illiterate | 11 (16.7) | 6 (60) | 4 (40) | 1.3(0.3–6.3) | 0.7 | - | - |
| $1^0$ school | 30 (45.5) | 19 (67.9) | 9 (32.1) | 0.9(0.3–3.1) | 0.9 | | |
| $2^0$ school/above | 23 (34.8) | 14 (66.7) | 7 (33.3) | 1 | | | |
| **Marital status** * | | | | | | | |
| Single | 53 (80.3) | 35 (74.5) | 12(25.5) | - | - | - | - |
| Married | 11 (16.7) | 2 (20) | 8 (100) | | | | |
| Divorced | 2 (3) | 2 (100) | 0 (0) | | | | |
| **Experience** | | | | | | | |
| ≤ 1 year | 43 (65.2) | 27 (73) | 10 (27) | 0.4(0.1–1.3) | 0.15 | 0.4(0.1–1.3) | 0.14 |
| > 1 year | 23 (34.8) | 12 (54.5) | 10(45.5) | 1 | | | |
| **Job division** | | | | | | | |
| Food preparation | 54 (81.8) | 9 (81.8) | 2 (18.2) | 0.3(0.1–1.9) | 0.2 | 0.3(0.1–1.9) | 0.23 |
| Cleaning utensils | 12 (18.2) | 30 (62.5) | 18(37.5) | 1 | | | |
| **Certified** * | | | | | | | |
| Yes | 2 (3) | 0 (0) | 2 (100) | - | - | - | - |
| No | 64 (97) | 39 (68.4) | 18(31.6) | | | | |

COR: Crude odds ratio, AOR: Adjusted odds ratio, CI: Confidence interval, 1: referent,

*: not tested for the association because one of the categories has a zero value: $N_1$ = Total number of participants interviewed; $N_2$ = total number of participants who gave stool sample

preparing food. Besides, 45.5% of the respondents claimed that they always wear hair restraints during food handling. Unfortunately, all of the participants claimed that they always handle food without glove. We also noted that 29 (43.9%) of the food handlers did not cut their fingernails at the time of assessment.

More than half (40/66) of the participants had good level of knowledge about foodborne diseases and food safety. The vast majority (93.9%) of them had ever heard of foodborne diseases. Likewise, 91% of the participants mentioned *Entamoeba histolytica* as a problem in food safety followed by *Giardia lamblia* (30, 45.5%). Meanwhile, 62 (93.9%) of the participants said that dysentery can be spread by contaminated food. By contrast, 65 of the 66 food handlers did not know that either Salmonella or Shigella causes foodborne disease. The study also found that 72.7% of the study participants did not assume that microbes can be found on skin of asymptomatic food handlers. More badly, 84.8% of the participants thought that contaminated food always shows some change in color, smell, or taste.

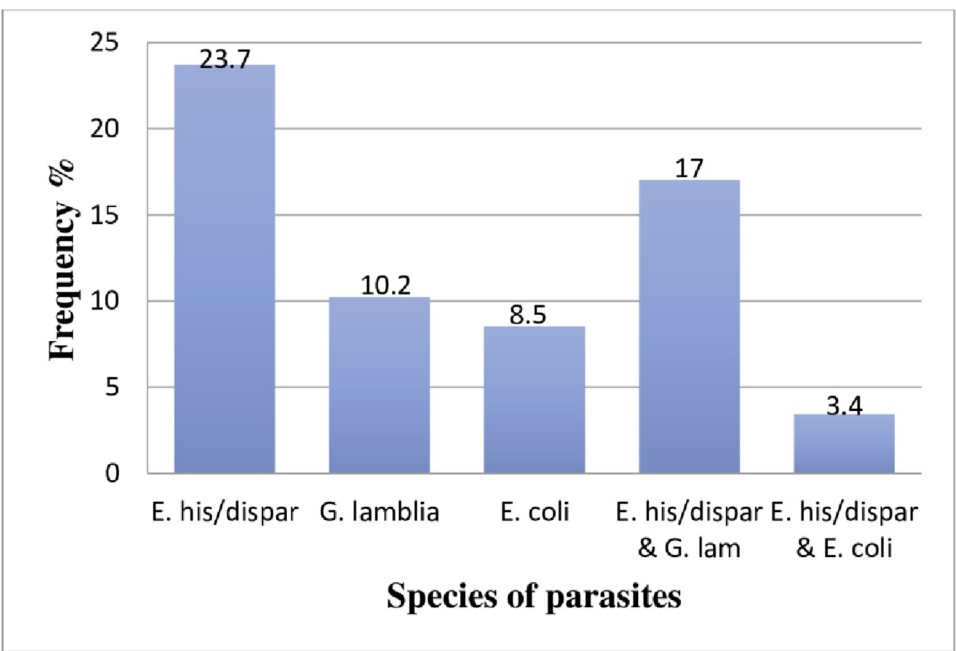

**Fig 1. The frequency of intestinal parasite species among the participants who gave stool specimens at Adigrat and Wukro prison centers, Tigrai, Northern Ethiopia, 2019.**

## Discussion

Foodborne diseases have long been affecting the health and the economic wellbeing of humans. Food handlers play significant roles in the transmission of these diseases in different communities [17]. There is limitation of published data on the knowledge and practice of food handlers toward foodborne diseases at prison institutions in Ethiopia, particularly in Tigrai region.

In our study, 37 of the 59 samples examined (62.7%) were positive for one or more intestinal parasites. This finding is comparable with other studies conducted in Ethiopia, 61.9% in Gojjam [15], and 52.1% in Nekemte town [22]. However, it is much higher than the report from Ethiopia (Axum town, 14%) [12] and Western Iran (9%) [23]. The variations in the

**Table 2. Antimicrobial susceptibility testing of Salmonella and Shigella isolates among the food handlers in Eastern Tigrai prison centers, Northern Ethiopia, 2019 (n = 4).**

| Drugs tested | Salmonella isolates (n = 2) | | Shigella isolates (n = 2) | |
|---|---|---|---|---|
| | Sensitive | Resistant | Sensitive | Resistant |
| | n (%) | n (%) | n (%) | n (%) |
| Gentamicin | 2 (100) | 0 (0) | 2 (100) | 0 (0) |
| Ciprofloxacin | 2 (100) | 0 (0) | 2 (100) | 0 (0) |
| Ceftriaxone | 2 (100) | 0 (0) | 2 (100) | 0 (0) |
| Amoxicillin | 0 (0) | 2 (100) | 0 (0) | 2 (100) |
| Tetracycline | 1 (50) | 1 (50) | 0 (0) | 2 (100) |
| Ampicillin | 0 (0) | 2 (100) | 1 (50) | 1 (50) |
| Chloramphenicol | 1 (50) | 1 (50) | 0 (0) | 2 (100) |
| Amoxicillin/clavulanic acid | 0 (0) | 2 (100) | 0 (0) | 2 (100) |
| Clarithromycin | 2 (100) | 0 (0) | 0 (0) | 2 (100) |

**Table 3. The knowledge and the practice of food handlers on foodborne diseases and food safety at Adigrat and Wukro prison centers, Northern Ethiopia (n = 66).**

| Questions regarding food handling practice | Response, n (%) | |
|---|---|---|
| | Yes | No/Don't know |
| Do you always wear gown while handling food? | 31 (47) | 35 (53) |
| Do you always wear hair restraint while handling food? | 30 (45.5) | 36 (54.5) |
| Do you always wear finger ornaments while preparing food? | 10 (15.2) | 56 (84.8) |
| Do you use gloves to prepare or handle prepared food? | 0 (0) | 100 (100) |
| Do you properly cover prepared food until consumption? | 22 (33.3) | 44 (66.7) |
| Have you ever prepared food while you have diarrhea? | 9 (13.6) | 57 (86.4) |
| Do you always wash food utensils just before use? | 34 (51.5) | 32 (48.5) |
| Do you always wash hands just before touching food? | 39 (59.1) | 27 (40.9) |
| Do you always wash your hands with soap and water after using toilet? | 53 (80.3) | 13 (19.7) |
| Do you always wash your hands after touching dirty material? | 54 (81.8) | 12 (18.2) |
| Was the hand fingernail of the food handler trimmed? | 37 (56.1) | 29 (43.9) |
| **Questions on the knowledge of food handlers on food safety** | **Yes** | **No/Don't know** |
| Have you ever heard of foodborne diseases? | 62 (93.9) | 4 (6.1) |
| Which species can cause foodborne disease? | | |
| Salmonella typhi | 1 (1.5) | 65 (98.5) |
| Shigella dysentriae | 1 (1.5) | 65 (98.5) |
| Entamoeba histolytica | 60 (91) | 6 (9) |
| Giardia lamblia | 30 (45.5) | 36 (54.5) |
| Do not know | 4 (6.1) | 62 (93.9) |
| Can an infected food handler transmit pathogens to the consumers? | 60 (91) | 6 (9) |
| Can washing hands before food contact reduce food contamination? | 100 (100) | 0 (0) |
| Can using glove to handle food reduces the risk of food contamination? | 2 (3) | 64 (97) |
| Can properly washing of utensils reduce the risk of food contamination? | 62 (93.9) | 4 (6.1) |
| Can typhoid fever be transmitted by contaminated food? | 32 (48.5) | 34 (51.5) |
| Can dysentery be spread by contaminated food? | 62 (93.9) | 4 (6.1) |
| Can microbes be found on the skin of asymptomatic food handlers? | 18 (27.3) | 48 (72.7) |
| Does contaminated food always show change in color, smell, or taste? | 56 (84.8) | 10 (15.2) |
| Should asymptomatic food handlers be evaluated during employment? | 61 (92.4) | 5 (7.6) |
| Can rodents/vectors spread foodborne pathogens? | 62 (93.9) | 4 (6.1) |

prevalence of intestinal parasites might be due to the differences in the methods of laboratory diagnosis applied, the environmental conditions, and most importantly the sample size. It appears likely that the mere presence of intestinal parasites among the food handlers in our study indicates inadequate hygiene.

We noted that *E. histolytica/dispar* was the dominant parasite detected among 23.7% of the participants who gave stool specimens. This is in keeping with other studies conducted in Ethiopia which reported a highest occurrence of this parasite [24, 25]. *E. histolytica* is among the common protozoan parasites, together with *Giardia lamblia* and *Cryptosporidium parvum*, which cause gastroenteritis in humans [26]. It is mainly transmitted via feco-oral route, with contaminated hands acting as major contributors to its transmission in areas where poor hygiene practices are common [27].

Of the 59 stools tested with culture and biochemical tests, 6.8% harbored either Salmonella or Shigella species (3.4% for each species). This finding lends support to previous studies that revealed 5.9% [13], and 5.04% [8] prevalence of these bacterial isolates among food handlers. Our finding was, however, lower than another study that reported a prevalence of 11.3% in

Gondar town, Ethiopia [28]. On the contrary, studies conducted in Jordan [29] and Iran [30] revealed neither Salmonella nor Shigella isolates among food handlers. These discrepancies may be attributed to the differences in the hygiene practices of the food handlers, the types of samples investigated, the socio-economic and education level of the participants. Food handlers have social responsibility to ensure food safety as 20% of foodborne diseases are transmitted due to improper food handling by food handlers [31]. This is especially common in prisons where prisoners live in overcrowded conditions and the health services are inadequate, with possible foodborne disease outbreaks.

On drug susceptibility, the Shigella and the Salmonella isolates were 100% sensitive to Gentamicin, Ciprofloxacin, and Ceftriaxone. Other similar studies have showed higher susceptibility of these bacteria to these antibiotics [13, 28]. Conversely, both the Shigella isolates were resistant to Amoxicillin, Tetracycline, Chloramphenicol, Amoxicillin/clavulanic acid, Clarithromycin, and 50% were resistant to Ampicillin. This was comparable with a study conducted at Haramaya University, Ethiopia that reported resistance of Shigella isolates to Tetracycline (76.2%), Amoxicillin (71.4%), and Chloramphenicol (66.7%) [8]. Additionally, a research from Arbaminch University, Ethiopia found that the Shigella isolates were 100% resistant to Amoxicillin and Clarithromycin and 40% to Amoxicillin/clavulanic acid [32].

The Salmonella isolates were also resistant to many of the antibiotics in the present study: 100% to Amoxicillin, Ampicillin, Amoxicillin/clavulanic acid, 50% to Tetracycline and Chloramphenicol. These findings suggested that the issue of antimicrobial resistance has not yet been resolved in Ethiopia. Antibiotic resistance has become a major public health threat throughout the globe, which requires collaborative intervention [33]. The main factors that contribute to antimicrobial resistances include mutations in bacterial genomes, inappropriate use of antibiotics, poor drug regulation policies, improper drug prescription, and disobedience to prescription [34].

Our data suggest that 51.5% and 60.6% of the participants had good level of practice and good level of knowledge, respectively. The level of practice of the study participants in our study is in keeping with a study conducted in Dangila town, Ethiopia; 52.5% had good level of practice [35]. Nevertheless, it is higher than the level of practice of the participants in Debark town (40.1%), Ethiopia [36] and much lower than a study in Jordanian military hospitals (89.4%) [37]. The differences in the level of practices among the different studies may be due to the variations in the method of evaluations, the level of knowledge of the participants, the working environments, or the socio-economic profiles of the participants.

In this study, 60.6% of the participants had good level of knowledge. This agrees with a similar study in Sri Lanka, 59.6% [38]. The majority of the participants in our study (93.9%) had ever heard of foodborne diseases. This is in line with a study from Dangila town (88.9%), Ethiopia [35], but contradicted with a study in India where 72.09% of the participants had never heard of foodborne diseases [39]. However, most of the participants (98.5%) in our study could not mention either Salmonella or Shigella species as problems in food. Similar to this, in one study [17], 76.2% of the study participants did not know that Salmonella causes foodborne infection. Besides, most of our participants (84.8%) assumed that contaminated food always shows some change in color, smell, taste. And surprisingly, only 3% (2/66) of the food handlers responded that handling food with gloves could reduce the risk of food contamination. This contradicts with a previous study [17], in which 77.9% of participants knew the importance of gloves in food handling. Our findings have pointed out that the food handlers in the study areas need health education or training programs on food safety and on the common pathogens that cause foodborne diseases.

During the study time, none of the participants used gloves, 53% did not wear gown, and 54.5% did not wear hair restraints while preparing food. The reasons, according to the

participants, were failure of the institutions to provide these materials. The WHO recommends wearing white coats during preparing and serving food to ensure that food is not exposed to any clothes worn underneath. Individuals engaged in food handling are also supposed to wear white caps or aprons to protect the food from hair [40].

## Limitations

The Salmonella and Shigella isolates in our study were identified only on species-level due to shortage in materials. Molecular techniques such as Polymerase Chain Reaction (PCR) were also unavailable to differentiate the Entamoeba complex species (*E. histolytica* and *E. dispar*) detected among the participants.

## Conclusions

In conclusion, foodborne pathogens are significant health problems in the study areas. Additionally, continuing health education and training programs are crucial to improve the level of knowledge and the level of practice of food handlers at the prison institutions. Provision of necessary facilities such as safe water supply, clean toilet, and soap is also recommended to enhance the personal hygiene of the food handlers.

## Supporting information

**S1 File.**
(ZIP)

## Acknowledgments

We are so grateful to Adigrat and Wukro prison center administrations and Clinic staff, Adigrat University, the data collectors, and participants for their valuable contribution to this work.

## Author Contributions

**Conceptualization:** Fitsum Mardu.

**Data curation:** Fitsum Mardu, Hadush Negash.

**Formal analysis:** Fitsum Mardu, Hadush Negash.

**Funding acquisition:** Fitsum Mardu.

**Investigation:** Fitsum Mardu, Haftom Legese, Kebede Tesfay.

**Methodology:** Fitsum Mardu, Hadush Negash.

**Project administration:** Fitsum Mardu.

**Resources:** Fitsum Mardu.

**Software:** Fitsum Mardu.

**Supervision:** Fitsum Mardu.

**Validation:** Fitsum Mardu, Hadush Negash.

**Visualization:** Fitsum Mardu.

**Writing – original draft:** Fitsum Mardu, Hadush Negash, Haftom Legese, Brhane Berhe, Kebede Tesfay, Hagos Haileslasie.

**Writing – review & editing:** Fitsum Mardu, Hadush Negash, Haftom Legese, Brhane Berhe, Kebede Tesfay, Hagos Haileslasie, Brhane Tesfanchal, Gebremedhin Gebremichail, Getachew Belay, Haftay Gebremedhin.

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
