## [Decision Letter · Decision Letter 0]

7 Aug 2020

PONE-D-20-14571

Assessment of knowledge, practice, and status of food handlers toward Salmonella, Shigella, and intestinal parasites: A cross-sectional study in Tigrai prison centers, Ethiopia, 2019

PLOS ONE

Dear Dr. landu,

Thank you for submitting your manuscript to PLOS ONE. After careful consideration, we feel that it has merit but does not fully meet PLOS ONE’s publication criteria as it currently stands. Therefore, we invite you to submit a revised version of the manuscript that addresses the points raised during the review process.

There are number of issues in methodology and manuscript presentation that need to be addressed.

We look forward to receiving your revised manuscript.

Kind regards,

Iddya Karunasagar

Academic Editor

PLOS ONE

Additional Editor Comments:

Three reviewers have commented on the manuscript and number of deficiencies have been pointed out. There are discrepancies in the number of samples tested and number stated in text and figures. Methodology used for identification of Salmonella is inadequate and the isolates can be termed "presumptive Salmonella". There are number of other comments made by the reviewers to address. Please revise the manuscript addressing all comments point by point.

Journal Requirements:

2. Thank you for stating in the text of your manuscript "Institutional Review Board (IRB) of Tigray Health Research Institute (THRI) has approved the study (THRI/4031/0393/2018). " Please also add this information to your ethics statement in the online submission form."

Please describe in your methods section how capacity to provide consent was determined for the participants in this study. Were there any actual or perceived negative impacts of not participating? How were those mitigated? Were the participants made aware of potential risks of participating or not participating in this study such as differential treatment?

Please also state whether your ethics committee or IRB specifically approved your consent procedure. If you did not assess capacity to consent please briefly outline why this was not necessary in this case.

Reviewers' comments:

Reviewer's Responses to Questions

**Comments to the Author**

1. Is the manuscript technically sound, and do the data support the conclusions?

Reviewer #1: Partly

Reviewer #2: Partly

Reviewer #3: Yes

2. Has the statistical analysis been performed appropriately and rigorously? 

Reviewer #1: Yes

Reviewer #2: I Don't Know

Reviewer #3: Yes

3. Have the authors made all data underlying the findings in their manuscript fully available?

Reviewer #1: Yes

Reviewer #2: Yes

Reviewer #3: Yes

4. Is the manuscript presented in an intelligible fashion and written in standard English?

Reviewer #1: No

Reviewer #2: Yes

Reviewer #3: Yes

5. Review Comments to the Author

Reviewer #1: The authors have conducted institutional based cross-sectional study among food handlers in prison centers. Although numbers of samples are less but the data generated is having public health significance. However, editing of language/ recasting of sentences are required to bring more clarity in the presentation. Length/size of table/presentation of data shall be reduced. Instead of giving all data/questionnaire as table, only selected /significant findings shall be given. There are many parameters/observations presented in tables 1and 3, which have not been discussed at all. Such insignificant observations may be deleted from tables.

At many places, authors have indicated that 66 volunteers participated in the study but 59 have given stool samples (eg. line 192) but at some places authors have indicated that 59 of the 66 samples examined were positive for one or more intestinal parasites (eg. line 236). This is confusing. Authors are requested to recheck and make necessary changes in calculation of percentages/presentation of data.

Similarly, in table 1 and table 3 and discussion, it is indicated that 66 participants participated in the study(eg line 182), but data given in column three (presence of foodborne pathogens) is not tallying with column 2.

Table 1, Gender, total male are 40, 20 are aware of foodborne pathogens but 13 are not?.

Similarly in other parameters eg Age in years, education level, marital status, experience, job division etc data is not tallying. In table 1, column 2, n is 66 but column 3 contains information related to only 59 respondents. Table 3 also contains responses from all 66 volunteers. There is need to bring clarity in presentation.

There were only two isolates for Salmonella and two for Shigella. It is not proper to indicate “all the isolates” (eg line 40, 203) of Salmonella or Shigella species were 100% or 50% sensitive/resistance for selected antibiotics. Instead of “two” isolates (line 208), words like both isolates/one isolate etc shall be used.

Lines 170- 179: require recasting to bring clarity in the presentation.

Table 1: Line 2, instead of using “n” twice ie (n,%), n=66, capital letter should be used for total number of samples (eg. N=66).

Table 2: Line 2, words like S, R, shall be elaborated (sensitive /resistant?) or given as foot note

Table 3, Line 20: Statement incomplete (Do you know……….).

Title shall be brief; year (2019) may be deleted.

Pl consult journal for presentation of references especially from Sl no. 1to 5, 7,10,14,18,19,20 and 40.

Thus there is need to recheck presentation of data and bring clarity in presentation/language before accepting for publication.

Reviewer #2: The study is carried out to assess the knowledge, practice, and magnitude of Salmonella, Shigella, and intestinal parasites among food handlers in Eastern Tigrai prison centers, Northern Ethiopia.

Food handlers have an important role in the spread of food-borne diseases. The personal hygiene and health status as well as their knowledge about various food-borne pathogens and good hygienic practices is important for prevention of spread of food-borne infections. A large number of food-borne outbreaks are reported from Ethiopia and the mortality due to these diseases is high. Therefore, the manuscript provides a valid rationale for the proposed studies, with clearly identified and justified research questions.

Authors have collected data using a structured questionnaire regarding the demographic characteristics, knowledge, and practice of participants on food safety and food-borne diseases. They have also analyzed stool samples for intestinal parasites and Salmonella and Shigella species from 66 food handlers, those outside the prison involved in the preparation of food and people inside the prison involved in food distribution.

The method followed for isolation and identification of Salmonella and Shigella is incomplete. Direct spreading of stool samples on Salmonella Shigella Agar, isolation of typical colonies from the plates and further biochemical identification can only result in presumptive identification of these pathogens. It is well proven by studies that a number of bacteria in stool samples or food or environmental samples can form typical colonies on Salmonella Shigella Agar, FDA BAM protocol, therefore, suggests use of three different selective media after prior enrichment in selective broth. Further, a number of these non Salmonella / Shigella cultures can give typical biochemical results. Therefore, it is necessary to carry out serological or molecular diagnostic tests to confirm the identity of these presumptive positive isolates. Therefore, the conclusion drawn only from biochemical identification about presence of Salmonella and Shigella in stool samples may be erroneous. Also, antimicrobial sensitivity studies with these presumptive isolates are not useful if these isolates are not pathogens claimed.

Authors are advised to test the presumptive positive isolates either by serology or by molecular diagnostic tests and resubmit the manuscript with the confirmatory test results.

Reviewer #3: comments are enclosed

Reviewer comments for manuscript entitled "Assessment of knowledge, practice, and status of food handlers toward Salmonella, Shigella, and intestinal parasites: A cross-sectional study in Tigrai prison centers, Ethiopia, 2019".

In this study, the authors have assessed the knowledge, practice and magnitude of foodborne pathogens like salmonella, shigella and intestinal parasites among food handlers. They had undertaken this study in Eastern Tigrai prison centres, Ethiopia. They have used structured questionnaires to collect demographic, characteristic, knowledge and practices of the participants on food safety with foodborne diseases. Authors observed that 62% of participants were found to harbour one or more intestinal parasites and nearly 7% were positive for one of the pathogens. However, Entamoeba spp was the most abundant parasite detected in nearly 25% of the population. Both the pathogen showed desirable antibiogram pattern. Authors also reported that 60% of participants had good knowledge of foodborne pathogens and nearly 52% were aware of food safety practices. The manuscript prepared by the authors had good study design, selection of area as well as study duration and enrollment of participant and had a good sample size and data collection. The standard procedure was followed for parasite identification and susceptibility testing. The standard statistical method was followed. Accordingly, the authors concluded that foodborne pathogens are the major concern among food handlers. Health, education and training programs are needed to improve the level of knowledge of food handlers. Other amenities like soap are recommended to enhance personal hygiene. The study is important in terms of food safety and hygiene and sufficient recommendations were made. The study done highlights the measures to be taken to educate and training of personnel. Hence, this manuscript is suitable for publication.

6. PLOS authors have the option to publish the peer review history of their article (what does this mean?). If published, this will include your full peer review and any attached files.

Reviewer #1: No

Reviewer #2: **Yes: **Dr. Jayant R. Bandekar

Reviewer #3: No

---

## [Author Response · Author response to Decision Letter 0]

16 Sep 2020

Reviewer #1: 

Comment 1: The authors have conducted institutional based cross-sectional study among food handlers in prison centers. Although numbers of samples are less but the data generated is having public health significance. However, editing of language/ recasting of sentences are required to bring more clarity in the presentation. Length/size of table/presentation of data shall be reduced. Instead of giving all data/questionnaire as table, only selected /significant findings shall be given. There are many parameters/observations presented in tables 1 and 3, which have not been discussed at all. Such insignificant observations may be deleted from tables.

Response: We believed that the paper needs revision in language and usage. Thus, we have revised the manuscript and corrected the language, grammar, punctuation, and sentence structures. Regarding the length/size of the data presented, most of the findings in the tables have not been discussed. But the authors still believe that these undiscussed findings are very important for readers. In fact, there are personal disagreements about the presentation of data in tables/figures and their description in text. However, we do not believe that findings should only be presented in tables unless to be discussed in text. Finally, if the reviewer believes that this is a critical revision, we are ready to reduce accordingly. 

Comment 2: At many places, authors have indicated that 66 volunteers participated in the study but 59 have given stool samples (e.g. line 192) but at some places, authors have indicated that 59 of the 66 samples examined were positive for one or more intestinal parasites (e.g. line 236). This is confusing. Authors are requested to recheck and make necessary changes in calculation of percentages/presentation of data.

Response: We thank the reviewer for this significant observation. As you can see in the results section, we stated that 37 of the 59 samples examined (62.7%) were positive for intestinal parasites. Now, we corrected the error (59 of the 66) in the discussion section. 

Comment 3: Similarly, in table 1 and table 3 and discussion, it is indicated that 66 participants participated in the study (e.g. line 182), but data given in column three (presence of foodborne pathogens) is not tallying with column 2.

Response: Here, the discrepancy is due to the difference in the number of participants interviewed (66) and those who provide stool samples (59). We included the 7 participants who completed the questionnaire but unable to provide samples to assess their knowledge and practice. Therefore, column 2 should tally 66 (total interviewed) and column 3 should tally 59 (total stool specimens examined). We now designated the number of participants interviewed (66) as ‘N1’ and the number of samples examined (59) as ‘N2’ in table 1. 

Comment: Table 1, Gender, total male are 40, 20 are aware of foodborne pathogens but 13 are not?. Similarly, in other parameters e.g. Age in years, education level, marital status, experience, job division etc. data is not tallying. In table 1, column 2, n is 66 but column 3 contains information related to only 59 respondents. Table 3 also contains responses from all 66 volunteers. There is need to bring clarity in presentation.

Response: By chance, out of the total 40 males interviewed, 7 of them did not provide stool samples. That is why total males in column 2 are 40 while in column 3 they are 33. Here, the 20 males (20/33) were positive for intestinal parasites. The same is true for age, education level …. Etc. Therefore, the confusion is due to the inclusion in the assessment of knowledge and practice of the 7 participants who did not provide stool samples but do complete the questionnaire. We believed assessing the knowledge and practice of these participants is very important even they failed to provide stool sample. 

comment: There were only two isolates for Salmonella and two for Shigella. It is not proper to indicate “all the isolates” (e.g. line 40, 203) of Salmonella or Shigella species were 100% or 50% sensitive/resistance for selected antibiotics. Instead of “two” isolates (line 208), words like both isolates/one isolate etc. shall be used. 

Response: We correct things accordingly.

Comment: Lines 170- 179: require recasting to bring clarity in the presentation. 

Response: We revised according to your comments. 

Comment: Table 1: Line 2, instead of using “n” twice i.e. (n, %), n=66, capital letter should be used for total number of samples (e.g. N=66).

Response: We designated total participants interviewed as N1= 66 and those who gave stool samples as N2= 59. 

Comment: Table 2: Line 2, words like S, R, shall be elaborated (sensitive /resistant?) or given as foot note

Response: Revised as commented

Comment: Table 3, Line 20: Statement incomplete (Do you know……….).

Response: It says ‘Do not know”. It is part of the preceding alternatives to the question “Which species can cause foodborne disease?”

Comment: Title shall be brief; year (2019) may be deleted.

Response: Year deleted from title. 

Pl consult journal for presentation of references especially from Sl no. 1 to 5, 7,10,14,18,19,20 and 40.

Thus there is need to recheck presentation of data and bring clarity in presentation/language before accepting for publication.

Reviewer #2: 

The study is carried out to assess the knowledge, practice, and magnitude of Salmonella, Shigella, and intestinal parasites among food handlers in Eastern Tigrai prison centers, Northern Ethiopia.

Food handlers have an important role in the spread of food-borne diseases. The personal hygiene and health status as well as their knowledge about various food-borne pathogens and good hygienic practices is important for prevention of spread of food-borne infections. A large number of food-borne outbreaks are reported from Ethiopia and the mortality due to these diseases is high. Therefore, the manuscript provides a valid rationale for the proposed studies, with clearly identified and justified research questions.

Authors have collected data using a structured questionnaire regarding the demographic characteristics, knowledge, and practice of participants on food safety and food-borne diseases. They have also analyzed stool samples for intestinal parasites and Salmonella and Shigella species from 66 food handlers, those outside the prison involved in the preparation of food and people inside the prison involved in food distribution.

comment: The method followed for isolation and identification of Salmonella and Shigella is incomplete. Direct spreading of stool samples on Salmonella Shigella Agar, isolation of typical colonies from the plates and further biochemical identification can only result in presumptive identification of these pathogens. It is well proven by studies that a number of bacteria in stool samples or food or environmental samples can form typical colonies on Salmonella Shigella Agar, FDA BAM protocol, therefore, suggests use of three different selective media after prior enrichment in selective broth. Further, a number of these non-Salmonella / Shigella cultures can give typical biochemical results. Therefore, it is necessary to carry out serological or molecular diagnostic tests to confirm the identity of these presumptive positive isolates. Therefore, the conclusion drawn only from biochemical identification about presence of Salmonella and Shigella in stool samples may be erroneous. Also, antimicrobial sensitivity studies with these presumptive isolates are not useful if these isolates are not pathogens claimed. Authors are advised to test the presumptive positive isolates either by serology or by molecular diagnostic tests and resubmit the manuscript with the confirmatory test results.

Response: We thank the reviewer for this important comment. The use of serological and molecular testing adjunct to culture and biochemical identification of salmonella and shigella isolates have significant relevance. However, as per the knowledge of the authors, presumptive isolates of salmonella and shigella by selective media can be confirmed by biochemical testing. In fact, there are some bacteria that show similar characteristics to salmonella or shigella species on individual biochemical tests. But we used four biochemical tests to exclude such phenotypic similarities on single biochemical tests. Also, there are many papers published on reputable scientific journals that have used similar techniques to identify salmonella and shigella species. But if the reviewer has strong evidence that it is impossible to do so, we are ready to rename the positive isolates as presumptive positive isolates and remove the antimicrobial susceptibility testing. This is because we cannot perform serology and molecular tests due to shortage of reagents and equipment. 

Reviewer #3: comments are enclosed 

Reviewer comments for manuscript entitled "Assessment of knowledge, practice, and status of food handlers toward Salmonella, Shigella, and intestinal parasites: A cross-sectional study in Tigrai prison centers, Ethiopia, 2019".

Comment: In this study, the authors have assessed the knowledge, practice and magnitude of foodborne pathogens like salmonella, shigella and intestinal parasites among food handlers. They had undertaken this study in Eastern Tigrai prison centres, Ethiopia. They have used structured questionnaires to collect demographic, characteristic, knowledge and practices of the participants on food safety with foodborne diseases. Authors observed that 62% of participants were found to harbour one or more intestinal parasites and nearly 7% were positive for one of the pathogens. However, Entamoeba spp was the most abundant parasite detected in nearly 25% of the population. Both the pathogen showed desirable antibiogram pattern. Authors also reported that 60% of participants had good knowledge of foodborne pathogens and nearly 52% were aware of food safety practices. The manuscript prepared by the authors had good study design, selection of area as well as study duration and enrollment of participant and had a good sample size and data collection. The standard procedure was followed for parasite identification and susceptibility testing. The standard statistical method was followed. Accordingly, the authors concluded that foodborne pathogens are the major concern among food handlers. Health, education and training programs are needed to improve the level of knowledge of food handlers. Other amenities like soap are recommended to enhance personal hygiene. The study is important in terms of food safety and hygiene and sufficient recommendations were made. The study done highlights the measures to be taken to educate and training of personnel. Hence, this manuscript is suitable for publication.

Response: We thank the reviewer very much

---

## [Editor Report · Decision Letter 1]

9 Oct 2020

Assessment of knowledge, practice, and status of food handlers toward Salmonella, Shigella, and intestinal parasites: A cross-sectional study in Tigrai prison centers, Ethiopia

PONE-D-20-14571R1

Dear Dr. landu,

We’re pleased to inform you that your manuscript has been judged scientifically suitable for publication and will be formally accepted for publication once it meets all outstanding technical requirements.

Kind regards,

Iddya Karunasagar

Academic Editor

PLOS ONE

Additional Editor Comments (optional):

All reviewer comments have been addressed.
---

## [Editor Report · Acceptance letter]

20 Oct 2020

PONE-D-20-14571R1 

Assessment of knowledge, practice, and status of food handlers toward Salmonella, Shigella, and intestinal parasites: A cross-sectional study in Tigrai prison centers, Ethiopia 

Dear Dr. landu:

I'm pleased to inform you that your manuscript has been deemed suitable for publication in PLOS ONE. Congratulations! Your manuscript is now with our production department. 

Kind regards, 

on behalf of

Dr. Iddya Karunasagar 

Academic Editor

PLOS ONE